# Sustaining compliance with hand hygiene when resources are low: A quality improvement report

Zaki Abou Mrad[1], Nicole Saliba[2], Dima Abou Merhi[2], Amal Rahi[2], Mona Nabulsi[2]*

1 Faculty of Arts and Sciences, Medical Research Volunteer Program (MRVP), American University of Beirut, Beirut, Lebanon, 2 Department of Pediatrics and Adolescent Medicine, American University of Beirut, Beirut, Lebanon

* mn04@aub.edu.lb

## Abstract

### Background

Sustainability of hand hygiene is challenging in low resource settings. Adding ownership and goal setting to the WHO-5 multimodal intervention may help sustain high compliance.

### Aim

To increase and sustain compliance of nursing and medical staff with hand hygiene in a tertiary referral center with limited resources.

### Methods

A quality improvement initiative was conducted over two years (2016–2018). After determining baseline compliance rates, the WHO-5 multimodal intervention was implemented with staff education and training, system change, hospital reminders, direct observation and feedback, and hospital safety climate. Additionally, the medical staff was responsible for continuous surveillance of compliance (*ownership*) until rates above 90% were achieved and sustained (*goal setting*).

### Results

Of 2987 observations collected between August 2016 and April 2018, 1630 (54.5%) were before, and 1357 (45.5%) were after patient encounters. The average overall compliance with hand hygiene was sustained at 94% for nursing and medical staff. Two instances of drops below 90% were associated with incidence of nosocomial *Rotavirus* infections. There were no similar infections during intervention periods with compliance rates above the set goal. Analysis using p-charts revealed significant improvement in compliance rates from baseline ($\chi^2$ (1) = 7.94, $p$ = 0.005).

**Data Availability Statement:** All relevant data are within the manuscript and its Supporting Information files.

**Funding:** The authors received no specific funding for this work.

**Competing interests:** The authors have declared that no competing interests exist.

## Conclusion

Adding *ownership* and *goal setting* to the WHO-5 multimodal intervention may help achieve, and sustain high rates of compliance with hand hygiene. Involving health care workers in quality improvement initiatives is feasible, durable, reliable, and cheap, especially in settings with limited financial resources.

## Introduction

Hospital-acquired infections have serious repercussions on patients' morbidity and mortality, length of hospital stay, and health care expenditure [1–3]. Adequate hand hygiene remains the most effective single preventive intervention [4, 5], which makes it the main focus of quality improvement programs in hospitals worldwide [6, 7]. However, compliance among health care workers averages about 50% [8], representing a major challenge to hospital-acquired infection control. Health care workers' compliance with hand hygiene is determined by knowledge and awareness about its importance, culture, memory and attention, and social influences [9]. Compliance with hand hygiene can be achieved with multimodal interventions such as the WHO-5 campaign, which has five components: system change, staff education and training, observation and feedback, hospital reminders, and hospital safety climate [10, 11]. Addition of goal setting, accountability, or reward incentives to WHO-5 may further increase compliance with hand hygiene [10].

This paper describes the journey of a pediatric department with limited human and financial resources to improve, and sustain health care workers' compliance with hand hygiene, and reduce *Rotavirus* hospital-acquired infections. Our quality improvement initiative was designed to increase and sustain compliance above 90% for at least two years. Our intervention was based on the WHO-5 model, and was implemented by a team of three members: the head of the pediatric quality unit, a departmental administrator, and a quality officer from the Hospital Quality Department. Medical staff monitored the intervention implementation. Our working hypothesis was that involvement of the medical staff will improve staff accountability, and sustain high compliance rates.

## Materials and methods

This project was mandated and approved by the Hospital Administration as a quality improvement initiative to address the reported *Rotavirus* nosocomial infections on the pediatric ward. Hence, it was exempt from review by the Institutional Review Board, and consent of patients and health care providers were not applicable in this case. This quality improvement initiative is reported in accordance with the Revised Standards for Quality Improvement Reporting Excellence (SQUIRE 2.0) [12].

### Context

Our center is a 344-bed tertiary care, university hospital, with 100 beds dedicated to the general pediatric ward, a pediatric intensive care unit, a neonatal intensive care unit, a normal newborn nursery, and a children's cancer center. The outpatient services provide primary and specialized care for 32,500 children each year. The Center has a well-established infrastructure for education, training and evaluation of staff in hand hygiene, and a well-advanced system change from this perspective. In 2015, the hospital Infection Control Office reported three

nosocomial *Rotavirus* infections, which coincided with hospitalization of children with community-acquired *Rotavirus* gastroenteritis that were attributed to breaches of hand hygiene before, or after patient encounters.

## Baseline data

Surveillance of physicians' compliance with hand hygiene before and after patient contact was conducted by student volunteers during November and December 2015 in all pediatric outpatient units. Thirteen students from the Medical Research Volunteer Program at our University were trained on administering a short survey to parents of children after their ambulatory visits to their physicians. The survey included four questions about patient satisfaction with the visit, and one question that asked whether the physician performed hand hygiene by washing hands with soap and water, or using a hand sanitizer before, or after contact with the child. The overall compliance rates were 97% (180/185) for before patient encounters, and 95.5% (172/182) for after patient encounters.

In January 2016 surveillance of the inpatient units was conducted. The nurse manager of each inpatient unit chose one nurse observer each day to conduct anonymous direct observations of physicians. One random observation was required for each physician daily. The physician's compliance with hand hygiene was observed during his/her patient round based on the WHO My 5 Moments of Hand Hygiene. The overall compliance was 92% (152/165) before, and 85.7% (84/98) after patient encounter.

To better understand the root causes of the low compliance rate among physicians working in the inpatient units, they were surveyed about their perceived barriers to strict hand hygiene practice, and their suggestions for improvement. The survey identified the following barriers: malfunctioning sinks, insufficient hand sanitizers at the point of care, inadequate distribution of sanitizers, and skin irritation from the disinfectant. Physicians suggested having reminders and flyers about My 5 Moments of Hand Hygiene next to patient care areas and sinks, more and better distribution of sanitizers, timely and periodic maintenance of sinks, and changing to a more skin-friendly sanitizer. The pediatric quality unit team rounded on all inpatient units, and inspected the functioning of the sinks, the availability of sufficient hand sanitizers (alcohol-based hand rub) at critical patient care areas like near patient bed and at room entrances, the availability of disposable gloves, disposable protective garments, and one stethoscope for each patient in critical care areas.

## Description of the intervention

In May 2016, the quality team decided on the WHO-5 plus as an intervention to improve hand hygiene compliance in the inpatient units, since baseline compliance rates in the outpatient units were above 90%. The intervention was a multi-component package that included staff education and training, system change, hospital reminders, direct observation and feedback, and hospital safety climate. Two other components were added: *ownership* and *goal setting* to sustain compliance above 90% in all inpatient units, for at least two years.

**Education and training.** The hospital's Infection Control Office delivered two educational sessions for all pediatric faculty and resident staff. These sessions aimed at developing awareness, and a culture of hand hygiene among all pediatric physicians. The sessions covered application of standard precautions, My 5 Moments of Hand Hygiene, the institutional Hand Hygiene Policy, the correct technique of hand rubbing and handwashing, and information about disinfectant properties. The pediatric quality team delivered user-centered education and training using a standardized approach for new trainees, and for physicians and nurses who were reported not to correctly perform hand hygiene.

**System change.** Nurse Managers of the pediatric inpatient units inspected hand rub availability at the point of care, and the functioning of sinks daily; and attended to deficiencies immediately. Flyers about My 5 Moments of Hand Hygiene in English and Arabic languages were placed on bulletin boards of nursing stations, next to sinks and hand rub bottles. Frequent reminders to physicians and nurses about hand hygiene were done daily by one champion nurse. Health care workers who reported sensitivity to the hand rub were very few (n = 3). They were instructed to wash their hands with soap and water before and after patient contact, instead of using the hand rub. The Chairperson of the department supported the hand hygiene intervention as the departmental performance improvement initiative for the year 2016, and that daily anonymous surveillance would be conducted, with provision of immediate feedback for non-compliers, and incentives for units with high compliance rates.

**Observation and feedback.** Starting August 2016, daily surveillance of hand hygiene was implemented in all pediatric inpatient units. In each unit, one champion from the unit's medical team was randomly chosen by the pediatric quality unit to conduct 20 anonymous direct observations over one week (average of three observations /day). The quality unit provided the champion with education and training about My 5 Moments of Hand Hygiene, and the correct way of hand rubbing and handwashing. Moreover, the champion was trained on how to record the observations on a data collection form developed for that purpose, and was instructed to email it to the quality officer as soon as the observation period was over. Non-compliers were sent emails by the Pediatric Quality Director about the details of the non-compliance incident, together with reminders about My 5 Moments of Hand Hygiene, and instructions about the correct technique of performing hand hygiene.

**Hospital safety climate.** Our hand hygiene quality improvement initiative was supported by the department chairperson, hospital administration, and the institutional Board of Trustees.

**Ownership and goal setting.** Ownership, our additional component to WHO-5 plus was implemented by having the medical staff be in charge of surveillance, and sharing with them compliance rates on quarterly basis. The set goal of achieving and sustaining compliance above 90% was disseminated to all health care workers in the inpatient units.

## Study of the intervention

Compliance rates were tracked on monthly basis by the Quality Director and the quality officer. Quarterly reports were generated for all, as well as individual inpatient units that included their overall, before and after patient compliance rates. Additionally, the Infection Control Program shared results of hospital-acquired *Rotavirus* infection surveillance with the Chairperson and Quality Director for cross-validation of data generated by the medical staff involved in the quality improvement initiative.

## Measures

Hand hygiene compliance rate was calculated as the number of compliant observations divided by the total number of observations (compliant and non-compliant). Observations collected by anonymous observers were reviewed by the quality officer, and checked for any missing or incomplete elements. When needed, the data were verified with the observer to ensure accuracy, and avoid false entries. Monthly data reports were generated and discussed with the Quality Director to monitor the progress of the intervention. Drops below the set goal of 90% were attributed to decreased compliance in a specific unit. They were promptly acted upon by the Quality Director by sharing the results with the Medical Director of the concerned unit, and discussion of potential causes that could explain the drop in compliance rates.

Moreover, actions to address hand hygiene breaches or system change were discussed and agreed upon when needed.

## Analysis

Quality control charts were used to monitor improvement in hand hygiene compliance over time. A p-chart was used to evaluate the overall compliance, as well as compliance in before and after patient encounters; nursing, and medical staff compliance. The upper and lower control limits (LCL) were set at 3ς, which equates to 3 standard deviations (SD). In the analysis, a run of eight consecutive points below or above the monthly average was considered a significant shift in the compliance rates ($p < 0.01$) [13]. In addition, a $\chi^2$ analysis was done to determine whether there was a significant change in hand hygiene compliance between the baseline data of 2016, and the post-intervention data of 2018.

## Ethical considerations

There were some ethical challenges encountered during implementation of the intervention. For example, few residents and medical students expressed concerns about reporting noncompliance of peers or faculty members, as this may create tension in the workplace. These concerns were addressed on one-to-one basis by the Quality Director. The concerned observers were reassured of the anonymity of their reports, and the importance of surveillance in promoting patient safety and quality care. Moreover, observers were offered the choice to continue, or decline participation in surveillance. Of 95 observer participants, only three opted to decline from participating in the surveillance. Another challenge was delayed observation submission due to the observers' busy schedules. Delayed submissions were excluded from analysis for fear of recall bias. Observers who were late to submit their observations were counseled about the importance of timely submission to ensure data validity. Only three observation forms were excluded over 21 months of surveillance. The third challenge was how to handle repeated noncompliance by the same individual, without having to resort to extreme disciplinary actions. The quality team opted to copy the supervisors of the concerned non-compliers on the email notifications of the specific incident, and how to avoid it in the future. The Quality Director met with non-compliant individuals after three violations to review their noncompliance, discuss barriers to proper hand hygiene, and highlight its impact on patients and on the hospital environment. Interestingly, none of the repeated offenders exceeded three noncompliance reports.

## Results

Between August 2016 and May 2018, a total of 2987 observations were collected: 1630 (54.5%) before, and 1357 (45.5%) after patient encounters. During the first year of the intervention (August 2016-August 2017), an average of 294 observations were conducted each month. The overall, as well as before patient encounter compliance were maintained above the 90% set goal (Figs 1 and 2).

After patient encounter compliance also reached to the set goal of 90% (Fig 3), except for three time points during which it dropped to 87% (November and December 2016, and August 2017). Despite these drops, the average compliance in the first year increased to 94% for all types of observations, as compared to baseline values (86% before, and 80% after patient encounter). Hence, the number of observations was reduced to an average of 57 per month starting August 2017.

During the second year (September 2017-May 2018), the overall, and after patient encounter compliance were sustained above 90% (Figs 1 and 3), with two drops in before patient

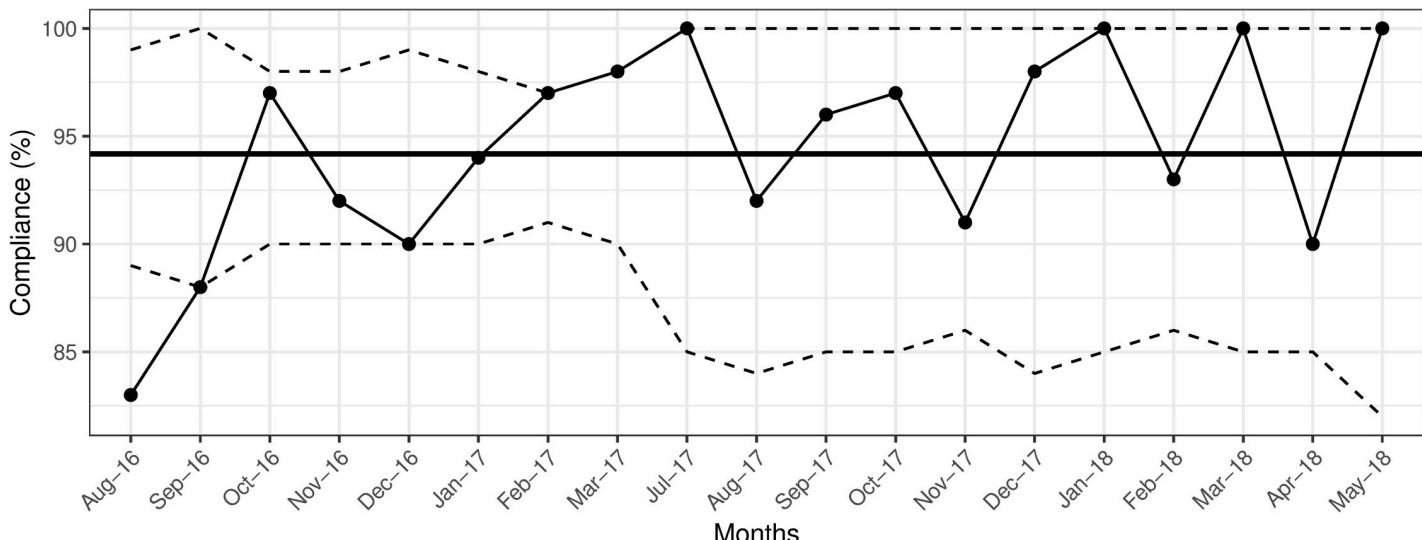

**Fig 1. Overall compliance with hand hygiene practice.** Solid bold line indicates the mean hand hygiene compliance. The dashed lines indicate upper and lower control limits, set at 3-ç.

encounter compliance reaching 88% in February and April 2018 (Fig 2). The average compliance during second year was 94% for both encounter types.

Analysis of compliance by health care worker discipline between November 2016 and May 2018 included 1067 observations on medical and 875 observations on nursing staff. Both disciplines achieved similar average compliance rates of 95%, for before, and after patient encounters. There was one drop in compliance to 85.2% for the nursing staff in November 2017 (Fig 4), the only drop below the LCL [13] after implementation of the intervention.

The overall compliance of medical staff also decreased to 84.3% during April 2018 (Fig 5).

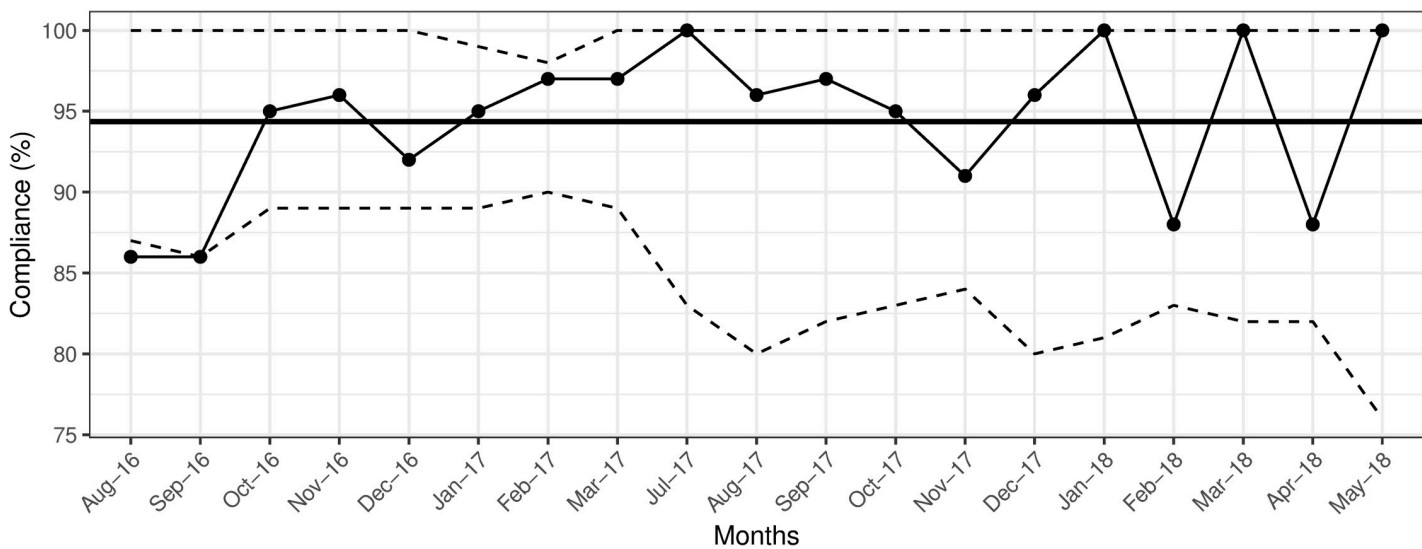

**Fig 2. Overall compliance with hand hygiene practice before patient encounters.** Solid bold line indicates the mean hand hygiene compliance. The dashed lines indicate upper and lower control limits, set at 3-ç.

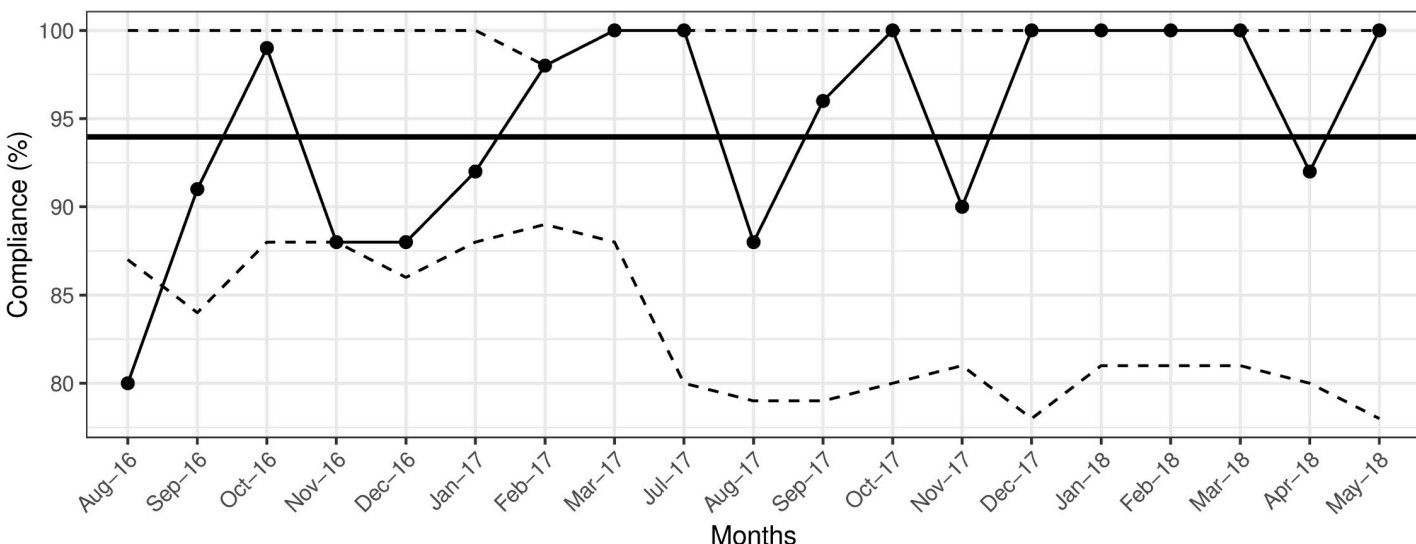

**Fig 3. Overall compliance with hand hygiene practice after patient encounters.** Solid bold line indicates the mean hand hygiene compliance. The dashed lines indicate upper and lower control limits, set at 3-ς.

Drops in compliance necessitated reminder sessions for physicians and nurses about proper hand hygiene. Cross-validation of the effectiveness of our intervention with the Infection Control Office data revealed no nosocomial *Rotavirus* infections during 2016 and 2018, and four such infections during 2017: two in January, one in May, and one in August, coinciding with drops in overall compliance with after patient encounters to levels at, or below the set goal of 90% (Fig 3). Except for the nursing staff data (Fig 4), p-charts revealed no significant shifts in data points above or below the mean following the implementation of the intervention in November 2016 (Figs 1–3 and 5). Hence, we considered the process to be *In Control* [13]. In

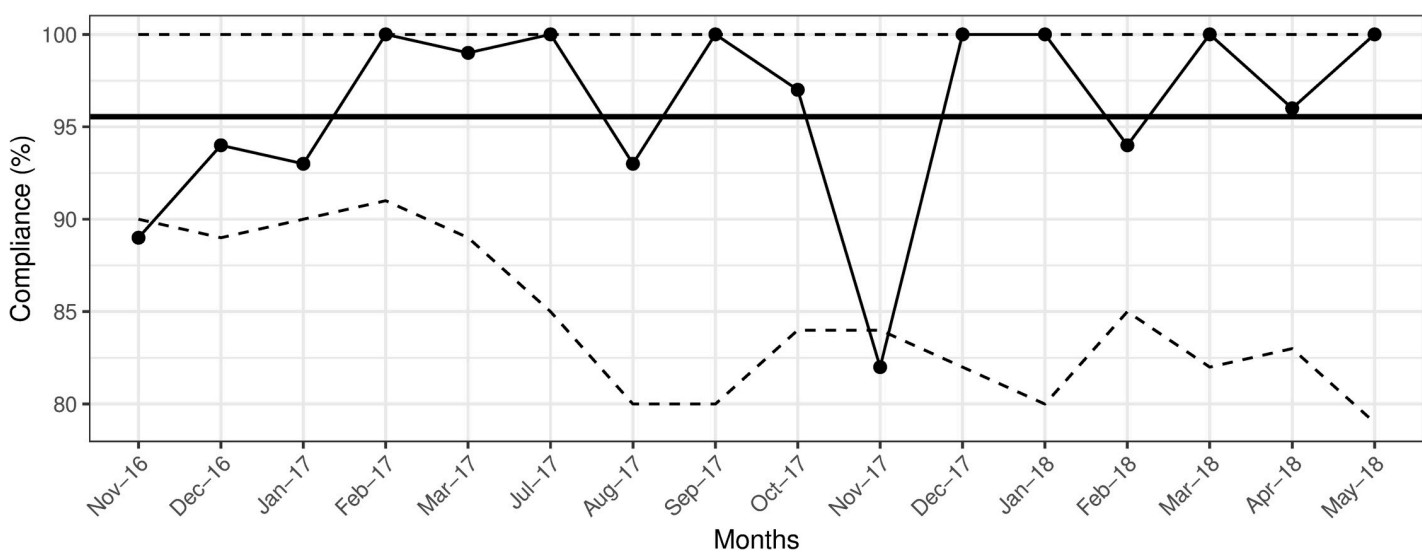

**Fig 4. Nursing staff overall compliance with hand hygiene practice.** Solid bold line indicates the mean hand hygiene compliance. The dashed lines indicate upper and lower control limits, set at 3-ς.

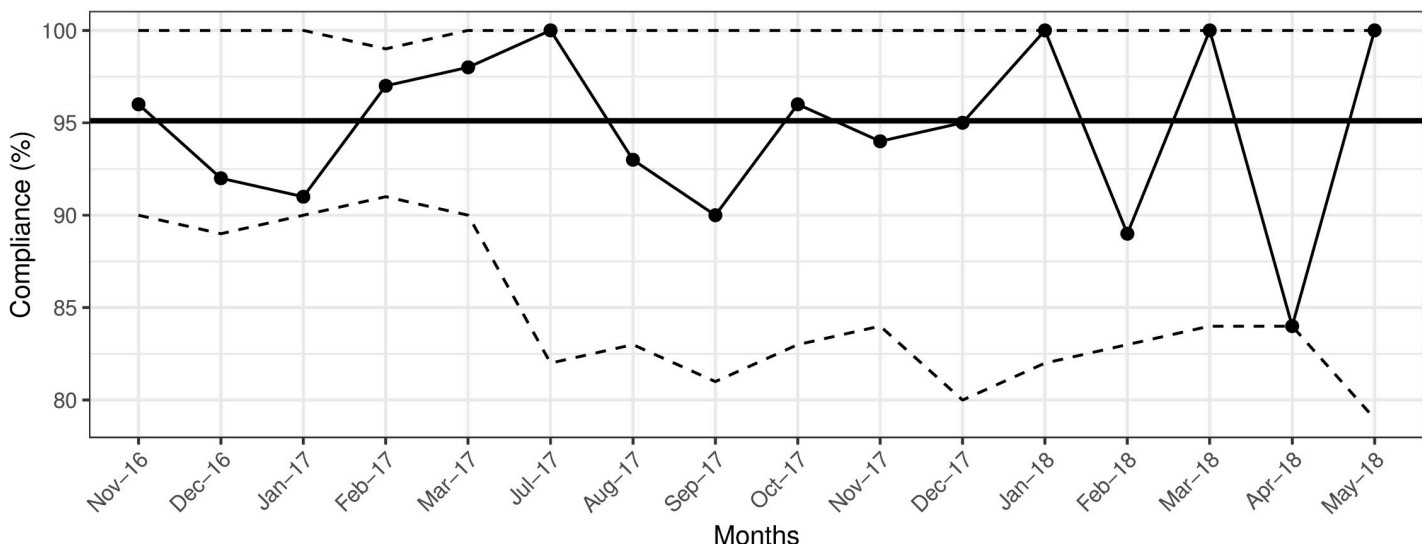

**Fig 5. Medical staff overall compliance with hand hygiene practice.** Solid bold line indicates the mean hand hygiene compliance. The dashed lines indicate upper and lower control limits, set at 3-ς.

addition, the $\chi^2$ test of independence suggested significant improvement in compliance rates between 2016 baseline data and 2018 data ($\chi^2$ (1) = 7.94, $p$ = 0.005).

## Discussion

This quality improvement initiative demonstrated that implementation of the WHO-5 multi-modal intervention in addition to *ownership* and *goal setting* increased and sustained hygiene compliance above the 90% set goal for twenty months in our inpatient units, a rate that is quite challenging to achieve and sustain. It has been shown that implementing accountability, reward incentives, or goal setting may help sustain high compliance rates [10, 14]. However, few studies reported on ownership of hand hygiene surveillance by physicians or nurses, all revealing a significant positive impact on compliance rates [15–19], The sustainability of high compliance rates in our setting may be attributed to the motivated physicians in charge of surveillance who may have felt that it was their responsibility to achieve the set goal. Physicians are the least involved in quality improvement initiatives, mainly because of time constraints and other work priorities. Hence recruiting them as internal champions may enhance success of quality improvement initiatives, especially if there is a supportive hospital leadership [16, 20]. Also, adding ownership to the WHO-5 multi-modal intervention did not mandate additional financial costs as it was integrated with the daily activity of our staff, making proper hand hygiene part of their daily routine, thus slowly changing their behavior.

We provided real-time performance feedback to non-compliers to raise their awareness of the importance of compliance for patient safety, a strategy that helped raise compliance after every drop, such as seen in November and December 2016, and August 2017. Several previous studies have shown that provision of real-time individual or group feedback was associated with improvement and sustainability of high compliance rates [21–24], as well as reduction in serious nosocomial infections such as central line-associated blood stream infections [25].

There are some limitations to our intervention. First, assigning surveillance ownership to health care workers would add additional tasks to their daily routine. This burden however can be reduced by increasing the pool of staff members participating in similar initiatives, as previously reported by Linam et al. [26]. Yet, it may be challenging to replicate this strategy in

small hospitals where human resources is an issue. Resorting to volunteers, such as medical students may be helpful in such settings. In our case, we trained volunteer medical students on proper hand hygiene and how to conduct direct observations of doctors and nurses. We found this experience to be useful since it saves on institutional human resources, is reliable, and sensitizes the students to the importance of proper hand hygiene for patient safety early on in their career. Rees et al. [27] and Ghee & Kowdley [17] also depended on volunteer medical students to implement their hand hygiene initiatives with similar success. A second limitation of our initiative is that some health care workers may be uncomfortable in reporting non-compliant colleagues or supervisors, especially in units with a small number of staff. Having video cameras for surveillance, instead of human observers can help avoid this concern. Automatic video monitoring has been used in auditing hand hygiene initiatives and providing real-time feedback to individual users [28]. Video surveillance can also reduce observer bias as reported by Sharma et al [29]. A third limitation is that sustainability of surveillance by hospital staff may be interrupted under certain circumstances, such as with new incoming staff, or periods in between academic years in teaching centers, which we faced during the month of June, and compelled us to temporarily withhold surveillance by the medical staff. This problem can also be overcome by using video monitoring. Finally, our data may be biased by the Hawthorne effect where doctors and nurses may have modified their behaviors because they knew they were being observed. However, cross-validation of our data against the hospital's Infection Control Office data on nosocomial *Rotavirus* infection revealed that the infections occurred with drops in after patient encounter compliance, suggesting that the Hawthorn effect on the validity of our data was minimal if any.

## Conclusions

Adding *ownership* and *goal setting* to the WHO-5 multimodal intervention may help achieve, and sustain high rates of compliance with hand hygiene. Involving health care workers in quality improvement initiatives is feasible, durable, reliable, and cheap. This strategy may be especially useful in settings with limited financial resources. Further study is needed to assess the feasibility of hospital-wide implementation of this strategy.

## Supporting information

**S1 Dataset. Anonymized pre-intervention data set.**
(XLSX)

**S2 Dataset. Anonymized post-intervention data set.**
(XLSX)

## Author Contributions

**Conceptualization:** Dima Abou Merhi, Amal Rahi, Mona Nabulsi.

**Data curation:** Zaki Abou Mrad, Nicole Saliba, Dima Abou Merhi, Mona Nabulsi.

**Formal analysis:** Zaki Abou Mrad, Nicole Saliba, Mona Nabulsi.

**Investigation:** Zaki Abou Mrad, Nicole Saliba, Dima Abou Merhi, Mona Nabulsi.

**Methodology:** Amal Rahi, Mona Nabulsi.

**Project administration:** Zaki Abou Mrad, Nicole Saliba, Mona Nabulsi.

**Supervision:** Nicole Saliba, Dima Abou Merhi, Mona Nabulsi.

**Validation:** Nicole Saliba, Mona Nabulsi.

**Visualization:** Amal Rahi, Mona Nabulsi.

**Writing – original draft:** Zaki Abou Mrad, Nicole Saliba, Dima Abou Merhi, Amal Rahi, Mona Nabulsi.

**Writing – review & editing:** Zaki Abou Mrad, Nicole Saliba, Dima Abou Merhi, Amal Rahi, Mona Nabulsi.

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
