## [Decision Letter · Decision Letter 0]

22 Jul 2020

PONE-D-20-06452

Sustaining compliance with hand hygiene when resources are low: a quality improvement report.

PLOS ONE

Dear Dr. Nabulsi,

Thank you for submitting your manuscript to PLOS ONE. After careful consideration, we feel that it has merit but does not fully meet PLOS ONE’s publication criteria as it currently stands. Two external reviewers evaluated your manuscript and highlighted a number of concerns regarding the study design and methodology; discussion of the findings; and the overall structure of the article. Therefore, we invite you to submit a revised version of the manuscript that addresses the points raised during the review process.

We look forward to receiving your revised manuscript.

Kind regards,

Joseph Donlan

Academic Editor

PLOS ONE

Journal Requirements:

Additional Editor Comments (if provided):

Regarding reviewer 1's point on ethical approval, we understand that you carried out a quality improvement study and that this was therefore exempt from the requirement for ethics committee approval, so there is no need to provide further justification on this aspect.

Reviewers' comments:

Reviewer's Responses to Questions

**Comments to the Author**

1. Is the manuscript technically sound, and do the data support the conclusions?

Reviewer #1: No

Reviewer #2: Yes

2. Has the statistical analysis been performed appropriately and rigorously? 

Reviewer #1: I Don't Know

Reviewer #2: Yes

3. Have the authors made all data underlying the findings in their manuscript fully available?

Reviewer #1: No

Reviewer #2: Yes

4. Is the manuscript presented in an intelligible fashion and written in standard English?

Reviewer #1: No

Reviewer #2: Yes

5. Review Comments to the Author

Reviewer #1: Dear authors, thank you for the opportunity to Review your article. I do have some Major concerns regarding your study:

1) There is no Ethical vote, though you stated that physicians and parents took part in a Survey.

2) Introduction line 59-60: I miss a Hygiene expert who performs the Training of the M5M of HH.

3) Methods: Your mehtods are a mix of methods and results. Furthermore, you only performed compliance measurements of two indications, mainly, before and after Patient contact. There are important HH indications missing. You mentioned a surveillance of physicians and parents, however, there are no results presented in dtail. In line 92 you mention low compliance rates, however, in the hole manuscript I do not find any very low compliance rate, they are rather high, in my eyes too high! You also mention 20 anonymous observations, how can Observation be like that when physicians receive Feedback after an Observation? Actually, I do not understand your methodological Approach, it is described in an unclear way. Line 159: Did you only collect data on before and after Patient contact?

4) Results: As you stated in two years 2987 observations were performded, which included only two indications! So patients never received an Infusion from a physician or nurses never had contact with Body fluid or the near Patient surrounding? Also the number of observations seems rather low. Did you also observe others (i.e.. dietology, physical therapy,..)? Overall, compliance rates seems rather high and 3 indications are missing. There are no surveillance data presented.

5) The discussion is poor and is lacking of recent literature (mostly between 2002 and 2013).

Kind regards

Reviewer #2: The manuscript provides an interesting insight into the implementation of the WHO-multimodal strategy to improve and sustain hand hygiene compliance rates in a setting with limited resources. Data provided are sound and analysis was performed in a rigorous way. However, some minor revisions are required.

INTRODUCTION

1. The authors reported that their quality improvement initiative was designed to increase and sustain compliance above 90% for at least two years. How did they identify such a threshold? Is there any evidence to identify as meaningful a cut-off of 90% and not below or above, like 85% or 95%? Please explain it.

METHODS

2. How did the nurse manager choose the observer each day? Alphabetic order, random order based on sequence, …? Besides, using such a high number of observers could lead to important differences. How was the inter-observer reliability tested? Were they trained? And how much time passed between conducting the observation and filling out the questionnaire? Was a recall bias possible?

3. “Non-compliers were sent emails by the Paediatric Quality Director about the details of the non-compliance incident, …”: does it mean that the observed healthcare worker’s name was disclosed in the observation form? Why?

RESULTS

4. Several papers in the literature have pointed out that hand hygiene compliance may vary across professional categories and in relation to several factors (i.e., shifts, types of interaction between patients and healthcare workers). Is any other information available from your study to support or dismiss these hypotheses?

DISCUSSION

5. “This quality improvement initiative demonstrated that adding ownership and goal setting to the WHO-5 multimodal intervention increased, and sustained hygiene compliance …”. I am not sure that this study really proved what the authors claim. We don’t know what would have happened if the interventions were carried out without these newly added components. It would be safer to say that the implementation of the whole strategy led to an increase in hand hygiene compliance rates.

6. In the baseline data, the authors report that “the overall compliance was 92% before, and 86% after patient encounter”. This result is worth a proper elaboration, as it seems to be in contrast with other findings where after patient’s contact the hand hygiene compliance rates were significantly lower, probably because hand hygiene was usually performed for self-protection.

7. I would say that real-time feedback to non-compliers is likely to be the main factor responsible for the successful campaign conducted in this hospital. Several other studies discuss the difficulty in maintaining a high rate of adherence to recommended practice over time and the importance of providing educational reinforcement and performance feedback to HCWs so that improvements can be sustained (for instance, see: Baccolini V, D'Egidio V, de Soccio P, Migliara G, Massimi A, et al. Effectiveness over time of a multimodal intervention to improve compliance with standard hygiene precautions in an intensive care unit of a large teaching hospital. Antimicrob Resist Infect Control. 2019; 8:92). I wonder what happened when the campaign stopped, and if would not have been appropriate to conduct a few observations after a few months. This could be added as an important limitation of the study.

6. PLOS authors have the option to publish the peer review history of their article (what does this mean?). If published, this will include your full peer review and any attached files.

Reviewer #1: No

Reviewer #2: No

---

## [Author Response · Author response to Decision Letter 0]

5 Sep 2020

Response to Reviewer #1

Thank you for taking the time to review our manuscript. We greatly appreciate your comments and suggestions. Kindly find below our response. 

1. There is no ethical vote, though you stated that physicians and parents took part in a survey.

Answer: The following paragraph is added to the beginning of the Methods section: “This project was mandated and approved by the Hospital Administration as a quality improvement initiative to address the reported Rotavirus nosocomial infections on the pediatric ward. Hence, it was exempt from review by the Institutional Review Board, and consent of patients and health care providers were not applicable in this case”.

2. Introduction, Line 59-60: I miss a hygiene expert who performs the training of the M5M of HH. 

Answer: The Departmental quality unit works closely with the Hospital Infection Control Office whenever such training is planned. We stated in the Education and training section that the training was done by the Infection Control Office: “The hospital’s Infection Control Office delivered two educational sessions for all pediatric faculty and resident staff. These sessions aimed at developing awareness, and a culture of hand hygiene among all pediatric physicians. The sessions covered application of standard precautions, My 5 Moments of Hand Hygiene, the institutional Hand Hygiene Policy, the correct technique of hand rubbing and handwashing, and information about disinfectant properties”. 

3. Methods: Your methods are a mix of methods and results. Furthermore, you only performed compliance measurements of 2 indications, mainly, before and after Patient contact. There are important HH indications missing. 

Answer: In reporting our Methods, we followed the Revised Standards for Quality Improvement Reporting Excellence (SQUIRE 2.0) [Reference 12]. Some subsections (such as Baseline data) require that we report the findings in this subsection rather than in the Results.

We agree with the kind reviewer that there are more HH indications than the before and after patient contact. However, for the purpose of this departmental quality improvement project we decided to focus on these 2 indicators only since they are by far the most commonly performed indications (as compared to procedures for example), and to minimize overwhelming our medical staff observers with less commonly performed indications that are routinely monitored/audited by the Infection Control Office. 

4. Methods: You mentioned a surveillance of physicians and parents, however, there are no results presented in detail. 

Answer: Yes. This surveillance was done in the pilot phase in the outpatient units to obtain baseline data. Since the overall before and after patient compliance in the outpatient units was above 90%, it was decided not to include it in the planned intervention since there was no problem there. We added the following sentence to summarize the results of the outpatient surveillance: “The overall compliance rates were 97% (180/185) for before patient encounters, and 95.5% (172/182) for after patient encounters”. As for the inpatient pre-intervention surveillance, we added the following sentence: “The overall compliance was 92% (152/165) before, and 85.7% (84/98) after patient encounter”. 

5. Methods: In line 92 you mention low compliance rates, however, in the whole manuscript I do not find any very low compliance rate. They are rather high, in my eyes too high! 

Answer: Yes. In the inpatient units, the before patient compliance was high at 92%, However since the after patient compliance of 86% was below our set goal of 90%, we decided to proceed with our intervention in the inpatient units. At the time this quality improvement was planned, the hospital was to undergo an assessment by the Joint Commission International (JCI) for accreditation. Hence, as part of the hospital preparation for this accreditation, the Administration asked each department to upgrade the standards of their quality indicators. In view of the Rotavirus nosocomial infections reported to us by the Infection Control Office, we elected to go for the set goal of 90% compliance. 

6. Methods: You also mention 20 anonymous observations, how can Observations be like that when physicians receive feedback after an observation? Actually, I do not understand your methodological approach. It is described in an unclear way. 

Answer: The identities of the physicians conducting the observations were not revealed to the observed staff (kept anonymous). However, because the multimodal intervention included ‘Feedback’ to non-compliers, we requested that the observers record only the name(s) of the non-complier(s), and the details of the specific violation(s) so we can give them feedback, as well as retrain them on the proper HH practice. 

7. Methods, Line 159: Did you only collect data on before and after Patient contact? 

Answer: Yes. We revised this sentence to become clearer for the reader: “Quarterly reports were generated for all, as well as individual inpatient units that included their overall, before and after patient compliance rates”.

8. Results: As you stated in 2 years 2987 observations were performed, which included only 2 indications! So patients never received an infusion from a physician or nurses never had contact with body fluid or the near patient surrounding? 

Answer: Patients did have a variety of procedures. However, as we mentioned in our reply to comment #3 above, for the purpose of this departmental quality improvement project we decided to focus only on these 2 indicators since they were by far the most commonly performed indications (as compared to procedures for example), and to minimize overwhelming our medical staff observers with less commonly performed indications that are routinely monitored/audited by the Infection Control Office. 

9. Results: Also the number of observations seems rather low. Did you also observe others (i.e. dietology, physical therapy..)? 

Answer: We specifically targeted the medical and nursing staff since it would be easier for the observers to conduct their observations in their teams. However there were very few observations done on other staff like those working in the cleaning services which are not included in this study. Luckily we had no violations among them. The Hospital Infection Control Office monitors/audits the HH practice of all kinds of hospital staff.

As for the number of the total observations being on the low side, we decided on 20 observations per week initially because we did not want to overwhelm the observers with too many observations that would interfere with their daily responsibilities, and might tempt them to withdraw from participation in the QI.

10. Overall, compliance rates seem rather high and 3 indications are missing. There are no surveillance data presented.

Answer: We addressed these issues in our reply above: for high compliance rates please see reply to comment #5, for the 3 missing indications please see reply to comments #3 and #8, and for the surveillance data please see reply to comment #4.

11. The discussion is poor and is lacking recent literature (mostly between 2002 and 2013). 

Answer: The Discussion section is now revised with the addition of 9 new recent references published between 2016 and 2020 (highlighted in yellow). 

Response to Reviewer #2

Thank you for your critical review of our manuscript and the valuable comments. Please find below our point by point response. 

1. Introduction: The authors reported that their quality improvement initiative was designed to increase and sustain compliance above 90% for at least 2 years. How did they identify such a threshold? Is there any evidence to identify as meaningful a cut-off of 90% and not below or above, like 85% or 95%? Please explain it.

Answer: When we designed this PI project, our literature review did not yield a specific recommended set goal by the CDC or the WHO. However, most of the papers in the literature used the 90% set goal (e.g. references #7, #17, #22; Omar et al. International Journal of Risk & Safety in Medicine. 2020 Jun 04; Linam et al. Pediatric Quality & Safety. 2017 Jul-Aug ;2(4):e035; and many papers from the older literature). Also, our Hospital Infection Control Office uses the same set goal. The Hospital’s rationale for this threshold was based on the fact that historically, the compliance in some of the hospital units was low ranging between 40% and 50%. Hence, to increase those pre-intervention low rates to rates high enough to secure patient safety, the 90% goal was chosen because it would almost double the low compliance rates and would be an achievable target, whereas the 100% compliance rate would be unachievable. 

2. Methods: How did the nurse manager choose the observer each day? Alphabetic order, random order based on sequence,..? Besides, using such a high number of observers could lead to important differences. How was the inter-observer reliability tested? Were they trained? And how much time passed between conducting the observation and filling out the questionnaire? Was a recall bias possible? 

Answer: The choice of the nurse observer in each unit was left to the nurse manager’s judgement based on the available nursing resources and the equity level of that unit. We did not interfere in their decisions or impose a specific system on them. Some units that had a good number of nurses chose the champion nurse from their pool of nurses who were never cited by the Infection Control Office to have had violations in hand hygiene practice (i.e. committed to good HH practice). The remaining units chose their champion nurse based on seniority and good audit training.

We did not do inter-observer reliability testing since the task to be observed was very simple, which is performing good hand rubbing with the hand sanitizer before and/or after patient contact. However, during training of all observers, each physician demonstrated to the quality officer how to properly perform good HH. Once this skill was mastered the physician was considered competent enough to recognize a HH violation when he/she observes one.

The observers were requested to record their observations as soon as they could do so in privacy, and before their shift was over. The possibility of recall bias of course cannot be completely ruled out, but since the shifts did not exceed 12 hours, this possibility is quite low. To note, log sheets that were submitted late were cancelled to avoid this bias. Only three observation forms were excluded over 21 months of surveillance. 

3. Methods: “Non-compliers were sent emails by the Pediatric Quality Director about the details of the non-compliance incident..”. Does it mean that the observed health care worker’s name was disclosed in the observation form? Why?

Answer: The identities of the physicians conducting the observations were not revealed to the observed staff (kept anonymous). However, because the multimodal intervention included ‘Feedback’ to non-compliers, we requested that the observers record only the name(s) of the non-complier(s), and the details of the specific violation(s) so we can give them specific feedback relating to their violation, as well as retrain them on the proper HH practice. There were no disciplinary measures. 

4. Results: Several papers in the literature have pointed out that hand hygiene compliance may vary across professional categories and in relation to several factors (i.e. shifts, types of interaction between patients and health care workers). Is any other information available from your study to support or dismiss these hypotheses?

Answer: We specifically targeted the medical and nursing staff since it would be easier for the observers to conduct their observations in their teams. However there were very few observations done on other staff like those working in the cleaning services. Luckily we had no violations among them, and these were excluded from this report. The reason for limiting the surveillance to nursing and medical staff is that we did not want to overwhelm the observers with too many observations as this would interfere with their daily responsibilities, and might tempt them to withdraw from participation in the QI. Our results revealed that medical and nursing staff had similar compliance rates, which is somehow different from the literature that report better HH compliance among nurses as compared to physicians. 

5. Discussion: “This quality improvement initiative demonstrated that adding ownership and goal setting to the WHO-5 multimodal intervention increased, and sustained hygiene compliance..”. I am not sure that this study really proved what the authors claim. We don’t know what have happened if the interventions were carried out without these newly added components. It would be safe to say that the implementation of the whole strategy led to an increase in hand hygiene compliance rates.

Answer: Thank you for the suggestion. We revised this sentence as follows: “This quality improvement initiative demonstrated that implementation of the WHO-5 multimodal intervention in addition to ownership and goal setting increased and sustained hygiene compliance above the 90% set goal..”.

6. Discussion: In the baseline data, the authors report that “the overall compliance was 92% before and 86% after patient encounter”. This result is worth a proper elaboration, as it seems to be in contrast with other findings where after patient’s contact the hand hygiene compliance rates were significantly lower, probably because hand hygiene was usually performed for self-protection.

Answer: We agree with the kind reviewer about this finding for which we have no explanation. We found only one report of a similar trend (Deshommes T, et al. A quality improvement initiative to increase hand hygiene awareness and compliance in a neonatal intensive care in Haiti. Journal of Tropical Pediatrics. 2020 June 28). In that paper, post-intervention hand hygiene was most likely to happen before, rather than after patient contact. We elected not to elaborate on why our pre-intervention after patient contact rate was lower than the before patient contact rate because we could not speculate about the reason, and because this trend did not persist post-intervention. We realize that this is opposite to what has been reported in the literature. 

7. Discussion: I would say that real-time feedback to non-compliers is likely to be the main factor responsible for the successful campaign conducted in this hospital. Several other studies discuss the difficulty in maintaining a high rate of adherence to recommended practice over time and the importance of providing educational reinforcement and performance feedback to HCWs so that improvements can be sustained (for instance, see Baccolini et al., 2019). I wonder what happened when the campaign stopped, and if it would not have been appropriate to conduct a few observations after a few months. This could be added as an important limitation of the study.

Answer: Yes, we agree with the kind reviewer that sustaining a high compliance rate is challenging. This is well-documented in the literature (thank you for the reference). In the provided reference (Baccolini et al., 2019), surveillance continued for 17 months only (November 2016- April 2018), whereas our surveillance continued for 21 months (August 2016- May 2018) in order to monitor sustainability. During the 21 months we continued to have high compliance rates which may be due to the real-time feedback to non-compliers, as per your suggestion. However, one cannot rule out the importance of other elements such as ownership which we believe is another major factor in the success of the intervention. In multi-component interventions, it is hard to dissect the individual contribution of each component. In June 2018 we stopped the HH surveillance by physician observers since the hospital was moving towards having electronic health record system using EPIC, and all hospital staff’s time (including physicians and nurses) was dedicated for training on EPIC. HH surveillance continued to be done by the Hospital’s Infection Control Office as usual. We were hoping to resume conducting our QI within 2 years however we were hit by the COVID-19 Pandemic similar to all other countries, which led to strict adherence to HH by all HCWs, patients and visitors. 

We hope we have addressed all the comments of the kind reviewers in a clear manner, and revised the manuscript accordingly.

Thank you again for the thorough review of our paper.

Yours sincerely, 

Mona Nabulsi, MD, MSc

Professor of Clinical Pediatrics

Department of Pediatrics and Adolescent Medicine

Faculty of Medicine

American University of Beirut

Beirut-Lebanon

P.O.Box: 113-6044/C8

Fax: 961-1-370781

 961-1-744464

E-mail: mn04@aub.edu.lb

---

## [Decision Letter · Decision Letter 1]

20 Oct 2020

Sustaining compliance with hand hygiene when resources are low: a quality improvement report.

PONE-D-20-06452R1

Dear Dr. Nabulsi,

We’re pleased to inform you that your manuscript has been judged scientifically suitable for publication and will be formally accepted for publication once it meets all outstanding technical requirements.

Kind regards,

Oathokwa Nkomazana, MD MSC PhD

Academic Editor

PLOS ONE

Additional Editor Comments (optional):

Reviewers' comments:

Reviewer's Responses to Questions

**Comments to the Author**

1. If the authors have adequately addressed your comments raised in a previous round of review and you feel that this manuscript is now acceptable for publication, you may indicate that here to bypass the “Comments to the Author” section, enter your conflict of interest statement in the “Confidential to Editor” section, and submit your "Accept" recommendation.

Reviewer #1: All comments have been addressed

Reviewer #2: All comments have been addressed

Reviewer #3: (No Response)

2. Is the manuscript technically sound, and do the data support the conclusions?

Reviewer #1: Partly

Reviewer #2: Yes

Reviewer #3: Yes

3. Has the statistical analysis been performed appropriately and rigorously? 

Reviewer #1: I Don't Know

Reviewer #2: Yes

Reviewer #3: Yes

4. Have the authors made all data underlying the findings in their manuscript fully available?

Reviewer #1: Yes

Reviewer #2: Yes

Reviewer #3: Yes

5. Is the manuscript presented in an intelligible fashion and written in standard English?

Reviewer #1: Yes

Reviewer #2: Yes

Reviewer #3: Yes

6. Review Comments to the Author

Reviewer #1: Dear authors, thank you very much for answering my comments. However, you discuss the WHO 5 multimodal model and Focus only on two indications. I cannot Support your opinion, that in a low resource Hospital there is no time for the other three indications. As this should represent a quality initiative I miss a comprehensive approach. It is the Patient which should be protected and it is the staff which should protect themselves. I am not convinced of this study at all.

Kind regards

Reviewer #2: all comments have been addressed. No further revision is required. The manuscript is clear and writtein in an intelligible fashion.

Reviewer #3: At the beginning of the method section you stated that the project was mandated and approved by the Hospital Administration, was exempt from IRB. This meant that participants had no choice. Why do you have a section on ethical considerations? Although strictly QI activities usually do not require IRB oversight, this particular QI initiative followed tenets of research, which include a hypothesis and some basic statistical analysis; as such an IRB review would have been appropriate. In addition, some ethical challenges were reported by few residents and medical students.The issue of anonymity even though addressed remains questionable; given that for non-compliers the quality team opted to copy their supervisors on emails which meant confidentiality between participants and the quality team was breached. Were there any penalties imposed on non-compliers?

7. PLOS authors have the option to publish the peer review history of their article (what does this mean?). If published, this will include your full peer review and any attached files.

Reviewer #1: No

Reviewer #2: No

Reviewer #3: No

---

## [Editor Report · Acceptance letter]

23 Oct 2020

PONE-D-20-06452R1 

Sustaining compliance with hand hygiene when resources are low: a quality improvement report 

Dear Dr. Nabulsi:

I'm pleased to inform you that your manuscript has been deemed suitable for publication in PLOS ONE. Congratulations! Your manuscript is now with our production department. 

Kind regards, 

on behalf of

Dr. Oathokwa Nkomazana 

Academic Editor

PLOS ONE